# Resonant Tunneling of Electrons and Holes through the In$_x$Ga$_{1-x}$N/GaN Parabolic Quantum Well/LED Structure

Hind Althib 

Department of Physics, College of Science, Imam Abdulrahman Bin Faisal University, P.O. Box 1982, Dammam 31441, Saudi Arabia; halthib@iau.edu.sa

**Abstract:** Models describing the tunneling of electrons and holes through parabolic In$_x$Ga$_{1-x}$N/GaN quantum well/LED structures with respect to strain were developed. The transmission coefficient, tunneling lifetime, and efficiency of LED structures were evaluated by solving the Schrödinger equation. The effects of the mole fraction on the structure strain, resonant tunneling and tunneling lifetime, and LH–HH splitting were characterized. The value of LH–HH splitting increased and remained higher than the Fermi energy; therefore, only the HH band was dominant in terms of the valence band properties. The results indicate that an increase in the mole fraction can lead to efficiency droop.

**Keywords:** InGaN/GaN; quantum resonant tunneling; parabolic quantum well/LED; heavy hole; light hole; multiple quantum wells; lifetime; internal quantum efficiency

## 1. Introduction

Since the early 1980s, superlattices with strained layers attracted a great deal of attention. By using ternary strained layer superlattices, it is possible to modify important material properties, such as the lattice constant, bandgap, and perpendicular transport [1]. Recent developments in optoelectronic and microelectronic device manufacturing have led to GaN and In$_x$Ga$_{1-x}$N/GaN becoming essential materials in the design of blue and green light-emitting diodes (LEDs). There are several advantages of GaN, such as its wide bandgap, high breakdown voltage, and excellent thermal and chemical resistance [2].

LED light emission efficiency has received significant attention for reducing electric power dissipation and increasing light output. To achieve this, many studies have tried to increase the carrier recombination rates and confine them within the active region. III-Nitride semiconductor materials (AlN, GaN, and InN) have been used as a substrate for light-emitting diodes (LEDs) [3,4]. Furthermore, in semiconductor systems, the phenomenon of the resonant tunneling of electrons through potential barriers has received much greater attention than the tunneling of holes in numerous theoretical and experimental investigations [5]. Due to their large effective mass and strong spin–orbit coupling, hole systems are of great importance [6].

Researchers developed a theoretical model through the comparison of experimental data, while simultaneously considering the tunneling and relaxation times of hole tunneling in a GaAs/AlGaAs double-quantum-well structure. In their study, they found that tunneling time increased as relaxation time decreased [7]. An empirical tight-binding model was used to calculate the transmission coefficient and current of the heavy hole and light hole band through the single and double barrier of GaAs/AlGaAs structures at different quantum barrier widths. They found that with a thicker quantum barrier, the peak numbers of heavy holes was less when compared to the peak numbers of light holes [5]. Likewise, for stepped In$_{0.2}$GaN$_{0.8}$/GaN quantum well-LED structures, the researchers experimentally determined that the thin barrier structure enhanced the device's performance and decreased efficiency droop at high current density. This may be caused by poor hole

tunneling through the thick barrier, causing electron leakage from the active region [8], while a theoretical study for parabolic $In_{0.2}GaN_{0.8}$/GaN quantum well-LED structures found that a thick barrier structure significantly increased efficiency [4].

According to a quantum tunneling theory, the electron energy increases with the decrease in the barrier height; therefore, it leads to the gradual decay of the electron wave. Conversely, the barrier height increases in the hole state, and the hole wave decays sharply [9]. A theoretical study examining the effects of plane wavevectors on the transmission coefficient of holes in GaMnAs/GaAlAs double-barrier structures was performed, wherein the plane wavevector tended to cause band mixing. With a zero plane wavevector, researchers found three peaks of transmission for HH → HH and one peak of transmission for LH → LH, which means that, in the quantum well, there are three quasi-bound states in the HH band and one quasi-bound state in the LH band. Moreover, the results demonstrate band mixing at the nonzero plane wavevector, with HH → LH and LH → HH and with HH → HH and LH → LH. Four resonant peaks appeared on all curves at the nonzero plane wavevector, corresponding to the four quasi-bound states in the QW [10]. Band mixing increases along with an increase in plane wavevector, quasi-bound state lifetime, and barrier width [11].

Hole–hole interaction effects at zero bias should be much stronger in quantum wire systems due to the enhanced hole effective mass, which becomes stronger with increasing temperature [6]. For nitride layers grown along the c-axis in a quantum well strain, the heavy hole (HH) and light hole (LH) bands are very close, i.e., the strain is not as effective as in GaAs- or InP-based lasers [12]. Here, the layered structures and the shapes of the multiple quantum wells play an important role. According to a theoretical study, finite parabolic quantum wells have a higher energy level than rectangular wells when comparing derived exact analytical solutions [12]. The barrier and quantum well layers were deposited using the molecular beam epitaxy (MBE) growth technique, whereby the barrier layer thickness quadratically increased with distance, while the quantum well layer thickness gradually decreased. A parabolic well is characterized by increased optical transitions, a greater resonance width, equally spaced energy levels, and the highest radiative recombination rate with uniform carrier distribution [13]. In addition, parabolic quantum wells have a strong sense of the energy level positions based on the valence band offset and their ability to confine potential [14]. Therefore, parabolic grading profile-MQW LEDs were found to have higher efficiency than stepped-MQW LEDs [15]. The effects of the InGaN/GaN quantum well LED structure shape on the hole transport in the active region have been studied. The results show that graded InGaN/GaN multiple quantum well/LED structures enhance hole transport in multiple quantum wells, even at low current density. Therefore, the current–voltage curve has lower series resistance than for stepped quantum wells, leading to a reduction in the efficiency droop of up to 70% at 20 $A/cm^2$ [16]. In one experimental study, the trapezoidal shape of InGaN/GaN multiple-quantum-well light-emitting diodes led to an increase in efficiency by enhancing the overlap of the electron and heavy hole wave functions at high current densities [17]. Similar results were reported in another study that improved the spatial overlap of the electron and hole wavefunctions in graded InGaN/GaN multiple-quantum-well LEDs [18]. In trapezoidal wells, however, there is a significant decrease in the separation between the electron and hole wave functions [18,19].

Experimental and simulation studies of the InGaN/GaN MQW LED structure revealed that the InGaN quantum well band becomes flat, and internal quantum efficiency (IQE) increases due to relaxation of the compressive strain in the GaN epilayer [20]. On the other hand, the relaxation of compressive stress in the GaN epilayers reduces piezoelectric polarization, which in turn reduces the quantum-confined Stark effect (QCSE). Polarization-induced QCSE limits the IQE of nitride LEDs [21]. Due to QCSE, the electron and hole wave functions partially separate, resulting in a reduction in radiative recombination and IQE [22]. Therefore, when reducing the QCSE, the light output power at a high current density is increased, and the efficiency droop is diminished. Although the increasing degree of relaxation leads to an increase in overlap of the electron and hole wave functions and

the probability of radiative recombination [23], the IQE may decrease due to the sharp decrease in hole concentration in the InGaN/GaN SQW and, therefore, a reduction in electron–hole pairs [20]. The researchers in an experimental study suppressed the negative impact of stress-induced QCSE via adopting a pre-well. Before growth of three pairs of In$_{0.01}$Ga$_{0.99}$N/GaN MQWs, five pairs of In$_{0.05}$Ga$_{0.95}$N/GaN superlattices, called pre-wells, were deposited on three pairs of superlattices [22].

Our study focused on investigating the influence of parabolic QW with respect to strain, and the effects of the mole fraction on the quantum resonant tunneling of electrons, heavy holes, and light holes through In$_x$GaN$_{1−x}$/GaN QW LED structures while theoretically determining their efficiency. A parabolic MQW LED structural model with an active region is proposed. The analysis considers the effect of variable effective mass in barrier and well layers. In addition, in this study, confluent hypergeometric functions were applied as part of the transfer-matrix method (TMM) to obtain exact solutions to the Schrödinger equation. The relationship between the transmission coefficient $\mathcal{T}$(E) and the electric field intensity of the structure was also investigated; the results were used to calculate the current density versus voltage (J–V) characteristics of the resonant tunneling diodes (RTDs) with a negative differential resistance (NDR). The relationship between the transmission coefficient $\mathcal{T}$(E) and the lifetime, with respect to the mole fraction of the structure, was also investigated. The Shockley equation was used to calculate the current density passing through the LED structure upon applying a voltage. The results of the current were used in the ABC model to determine the efficiency of the LED structure, and the results were compared with the experimental data [16].

## 2. Theoretical Model

Figure 1 shows the layered structures for the parabolic MQW LEDs employed in this study. The active region of the parabolic QW LED consisted of the following layers: GaN (1 nm), In 0 → 0.2 Ga1 → 0.8 N (1.75 nm), In 0.2 Ga 0.8 N (0.5 nm), In 0.2 → 0 Ga 0.8 → 1 N (1.75 nm), and GaN (1 nm). The potential energy of the parabolic MQW LED structure was changed under bias voltage. After annealing, the In concentration profile across InGaN/GaN as a function of the diffusion length (Ld) in the growth direction $z$ is given by Fick's law as follows [24]:

$$C(z) = \frac{1}{2}C_0 \left[ \text{erf}\left( \frac{\frac{W}{2} - z}{L_d} \right) - \text{erf}\left( \frac{\frac{W}{2} + z}{L_d} \right) \right], \tag{1}$$

where $C_0$ is the initial In concentration, and W is the well width.

The potentials for electrons, heavy holes, and light holes are given by

$$U_r = Q_r \left[ E_g(z) - S_{r\perp} \right] \pm S_{r\parallel}, \tag{2}$$

where $r$ indicates the electron, heavy hole (HH), and light hole (LH); $Q_r$ is the band offset splitting ratio, signs (+) and (−) indicate (HH) and (LH), respectively; and $S_{r\perp}$, and $S_{r\parallel}$ are the hydrostatic and shear strain, respectively [25].

$$S_{r\perp} = -a_v \left( \varepsilon_{xx} + \varepsilon_{yy} + \varepsilon_{zz} \right),$$

$$S_{r\parallel} = -\frac{b}{2} \left( \varepsilon_{xx} + \varepsilon_{yy} - \varepsilon_{zz} \right),$$

$$\varepsilon_{xx} = \varepsilon_{yy} = \frac{a_s - a_0}{a_0}, \text{ and } \varepsilon_{zz} = -2\frac{C_{12}}{C_{11}}\varepsilon_{xx},$$

where $a_v$ and $b$ are Bir–Pikus deformation potentials for the valence band, $a_s$ and $a_0$ are lattice constants of the substrate and layer materials, respectively, and $C_{11}$ and $C_{12}$ are elastic stiffness constants [26].

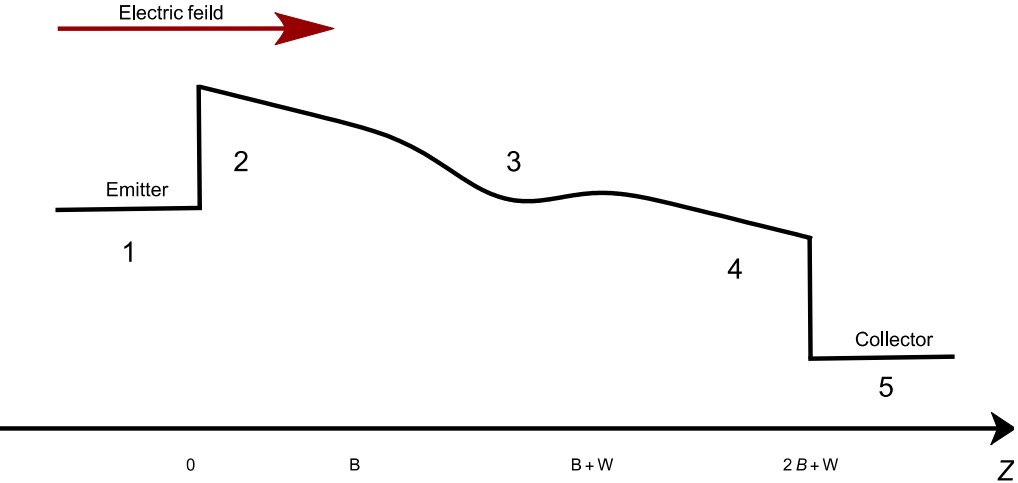

**Figure 1.** Schematic diagram of the InGaN/GaN parabolic QW LED structures under a uniform applied electric field.

$E_g(z)$ is the energy gap as a function of z, according to Vegard's law.

$$E_g(z) = x\,E_g(InN) + (1-x)E_g(GaN) - \beta\,x(1-x),\tag{3}$$

where $\beta$ is the bandgap bowing parameter.

The time-independent Schrödinger equation in a one-dimensional form and, in the z-direction for each layer of the parabolic MQW structure under a uniform applied electric field, is as follows [4]:

$$\left.\begin{array}{ll} -\frac{\hbar^2}{2m^*}\frac{\partial^2\psi_1(z)}{dz^2} - E\psi_1(z) = 0 & For\ 0 > z \\[4pt] -\frac{\hbar^2}{2m^*}\frac{\partial^2\psi_2(z)}{dz^2} + \left(V_0(z) - \frac{e\,V_a}{2B+W}z - E\right)\psi_2(z) = 0 & For\ z < B \\[4pt] -\frac{\hbar^2}{2m^*}\frac{\partial^2\psi_3(z)}{dz^2} + \left(V_0(z)\left(\frac{2z}{W}\right)^2 - \frac{e\,V_a}{2B+W}z - E\right)\psi_3(z) = 0 & For\ B \le z \le B+W \\[4pt] -\frac{\hbar^2}{2m^*}\frac{\partial^2\psi_4(z)}{dz^2} + \left(V_0(z) - \frac{e\,V_a}{2B+W}z - E\right)\psi_4(z) = 0 & For\ B+W < z \\[4pt] -\frac{\hbar^2}{2m^*}\frac{\partial^2\psi_5(z)}{dz^2} - (e\,V_a + E)\,\psi_5(z) = 0 & For\ z > 2B+W \end{array}\right\},\tag{4}$$

where $m^*$, $\hbar$, $E$, $V_0(z)$, and $\psi(z)$ are the effective mass, reduced Planck's constant, electron energy, potential energy, and wave function, respectively; $e$ is the electron charge ($e = 1.602 \times 10^{-19}C$); $V_a$ is the applied bias voltage; and $B$ and $W$ are the quantum barrier width and well width, respectively.

The conditions for the continuity of the wave function and its derivative at each interface $z = B$ and $z = B + W$ are as follows:

$$\left.\begin{array}{c} \psi_i(z)|_{z=(z_i)} = \psi_{i+1}(z)|_{z=(z_i)} \\[6pt] \frac{1}{m_i^*}\frac{d\psi_i(z)}{dz}\bigg|_{z=(z_i)} = \frac{1}{m_{i+1}^*}\frac{d\psi_{i+1}(z)}{dz}\bigg|_{z=(z_i)} \end{array}\right\},\tag{5}$$

where $i$ is the layer of the parabolic MQW structure.

The variables $\zeta(z) = \beta^{\frac{1}{3}}\left(\frac{2}{W}z + \frac{\alpha}{\beta}\right)$ are introduced, where

$$\alpha = \frac{2m_b^*\left(\frac{W}{2}\right)^2}{\hbar^2}(-V_0(z)+E),\quad \beta = \frac{2m_b^*\left(\frac{W}{2}\right)^3\frac{e\,V_a}{2\,B+W}}{\hbar^2},$$

$$\zeta(z) = \sqrt{2}\varrho^{\frac{1}{4}}\frac{2}{W}z - \frac{\sigma}{\sqrt{2}\varrho^{\frac{3}{4}}},\quad \epsilon(z) = \frac{\gamma}{\varrho^{\frac{1}{2}}} - \frac{\sigma^2}{4\varrho^{\frac{3}{2}}},$$

$$\gamma = \frac{2m_w^*\left(\frac{W}{2}\right)^2}{\hbar^2}E, \ \sigma = \frac{2m_w^*\left(\frac{W}{2}\right)^3 \frac{e\,V_a}{2\,B+W}}{\hbar^2},$$

$$\varrho = -\frac{2m_w^*\left(\frac{W}{2}\right)^2}{\hbar^2}V_0(z).$$

The solutions of Equation (4) are a linear combination of independent complex exponential functions and independent confluent hypergeometric functions [15,27–30].

The solutions are in the following form:

$$\left.\begin{aligned}
\psi_1(z) &= exp(jk_1z) + R\,exp(-jk_1z) \\[4pt]
\psi_2(z) &= C_2^+\,Ai(\xi\xi(z)) + C_2^-\,Bi(\xi\xi(z)) \\[4pt]
\psi_3(z) &= C_3^+\,\mathcal{M}_1(\zeta(z)) + C_3^-\,\mathcal{M}_2(\zeta(z)) \\[4pt]
\psi_4(z) &= C_4^+\,Ai(\xi\xi(z)) + C_4^-\,Bi(\xi\xi(z)) \\[4pt]
\psi_5(z) &= S\,exp(jk_5z)
\end{aligned}\right\}, \tag{6}$$

where $Ai$ and $Bi$ are linear combinations of Airy functions [15,28]. $\mathcal{M}_1(\zeta(z))$ and $\mathcal{M}_2(\zeta(z))$ are Weber functions [28], as shown below.

$$\mathcal{M}_1(\zeta(z)) = 2^{\frac{\epsilon(z)-1}{2}}exp\left(-\frac{(\zeta(z))^2}{4}\right)\frac{\sqrt{\pi}}{\Gamma\left(\frac{3-\epsilon(z)}{4}\right)}\mathcal{F}\left(-\frac{\epsilon(z)-1}{4},\frac{1}{2},\frac{1}{2}(\zeta(z))^2\right)$$

$$\mathcal{M}_2(\zeta(z)) = -2^{\frac{\epsilon(z)-1}{2}}exp\left(-\frac{(\zeta(z))^2}{4}\right)\frac{\sqrt{2\pi}\,\zeta(z)}{\Gamma\left(-\frac{\epsilon(z)-1}{4}\right)}\mathcal{F}\left(\frac{3-\epsilon(z)}{4},\frac{3}{2},\frac{1}{2}(\zeta(z))^2\right)$$

$\Gamma(\overline{x})$ is the gamma function, and $\mathcal{F}\left(\overline{a},\overline{b},\overline{x}\right)$ is the confluent hypergeometric function. Lastly, $j = \sqrt{-1}$, $k_1 = \sqrt{\frac{2m_w^*E}{\hbar^2}}$, and $k_5 = \sqrt{\frac{2m_w^*(E-V_w(z))}{\hbar^2}}$; $\{C_2^+, C_3^+, C_4^+\}$ and $\{C_2^-, C_3^-, C_4^-\}$ are the amplitudes of the running waves, indicating the incident wave and reflected wave, respectively [15,28]. In Regions 1 and 5, the amplitude of a plane wave traveling in the positive or negative $z$-direction is assumed to be $S$ and $R$ [29]; $m_w^*$, $m_b^*$, and $V_w(z)$ are the effective mass of the QW, effective mass of the quantum barrier, and QW potential, respectively.

The transmission coefficient can be calculated using the TMM [15]. Thus, the transmission coefficient for each of the five layers can be expressed as

$$\begin{bmatrix}1 \\ R\end{bmatrix} = \prod_{i=1}^{4}M_i\begin{bmatrix}S \\ 0\end{bmatrix} = \begin{bmatrix}T_{11} & T_{12} \\ T_{21} & T_{22}\end{bmatrix}\begin{bmatrix}S \\ 0\end{bmatrix}. \tag{7}$$

The fraction of incident particles transmitted by the barriers is given as follows [28]:

$$\mathcal{T}(E) = 1 - \frac{|T_{21}|^2}{|T_{11}|^2}, \tag{8}$$

where $T_{11}$, $T_{12}$, $T_{21}$, and $T_{22}$ are the matrix elements.

Substituting Equation (5) into Equation (6) yields a system of linear equations that can be represented by the matrices $M_i$.

The current density $J(E)$ through the resonant tunneling structure at a given bias voltage can be calculated by applying the transmission coefficient as a function of energy

$\mathcal{T}(E)$. The current density formula can be expressed using the Tsu–Esaki formula [31], as follows:

$$J = \frac{e \, m_w^* \, k_B \, \theta}{2 \, \pi^2 \, \hbar^3} \int_0^\infty \mathcal{T}(E) \, Ln \left( \frac{1 + \exp\left(\frac{E_f - E}{k_B \, \theta}\right)}{1 + \exp\left(\frac{E_f - E - e \, V_a}{k_B \, \theta}\right)} \right) dE \tag{9}$$

Then,

$$\left. \begin{array}{l} J = \frac{e \, m_w^*}{2 \, \pi^2 \, \hbar^3} \int_0^{E_f} \left( E_f - E \right) \mathcal{T}(E) dE, \ V_a \geq E_f \\[12pt] J = \frac{e \, m_w^*}{2 \, \pi^2 \, \hbar^3} \left[ V_a \int_0^{E_f - V_a} \mathcal{T}(E) dE + \int_{E_f - V_a}^{E_f} \left( E_f - E \right) \mathcal{T}(E) dE \right] V_a < E_f \end{array} \right\} \tag{10}$$

where $E_f$ is the Fermi level, $k_B$ is Boltzmann's constant, and $\theta$ is the absolute temperature.

For the measured I–V characterization, the current of a diode $I$ as a function of the forward voltage can be calculated by applying the Shockley equation, as follows [32]:

$$I = I_s \left[ \exp\left( \frac{e \, V_a}{k_B \, \theta} \right) - 1 \right], \tag{11}$$

where $I_s$ is the saturation current density, expressed as follows:

$$I_s = e \, \mathcal{A} \, n_i^2 \left( \frac{D_h}{L_h \, N_D} + \frac{D_e}{L_e \, N_A} \right), $$

where $\mathcal{A}$ is the diode area, $D$ and L are the diffusion coefficient and diffusion length for the electron and hole, and $n_i^2$ is the intrinsic concentration, expressed as follows:

$$n_i^2 = N_c \, N_v \exp\left( -\frac{E_g}{k_B \, \theta} \right), $$

where $E_g$ is the energy gap, and $N_c$ and $N_v$ are donors and acceptors, respectively.

The internal quantum efficiency (IQE) in $In_xGa_{1-x}N/GaN$ can be calculated using the ABC model, which can be written as follows [32,33]:

$$IQE = \frac{B \, N^2}{A \, N + B \, N^2 + C \, N^3}, \tag{12}$$

where $A$, $B$, $C$, and $N$ represent the Shockley–Read–Hall nonradiative recombination, bimolecular radiative recombination, Auger recombination, and carrier density, respectively. The recombination rate for carriers in a single QW device is given by

$$\frac{dN}{dt} = \frac{\eta \, J}{e \, d} - A \, N + B \, N^2 + C \, N^3, \tag{13}$$

where $\eta$ is the injection efficiency. Under the assumptions of steady state and $\eta = 1$, Equation (13) can be simplified as

$$\frac{J}{e \, d} = A \, N + B \, N^2 + C \, N^3. \tag{14}$$

Equation (14) can be solved as a linear cubic equation with one variable, $N$, which has two complex solutions and one real solution [4]. The IQE can be calculated by substituting the real solution $N$ into Equation (12).

## 3. Results and Discussion

Figure 2 presents a schematic band diagram of the InGaN/GaN parabolic quantum well with a diffusion length of 1.75 nm, showing the conduction band and the valence band. The valence band is degenerated into heavy and light hole bands (HH and LH,

respectively). Taking into account the type of strain, i.e., compressive strain, the HH band is higher than the LH band in the InGaN quantum well, as the effective mass of HH is $1.68915\ m_0 > 0.193942\ m_0$ (where $m_0$ is the electron mass). In this case, the energy gap is calculated by the energy difference between the bottom of the conduction band and the top of the HH band. Compressively strained profiles of the InGaN/GaN parabolic quantum well with various mole fractions ($x_0$) are shown in Figures 3–5.

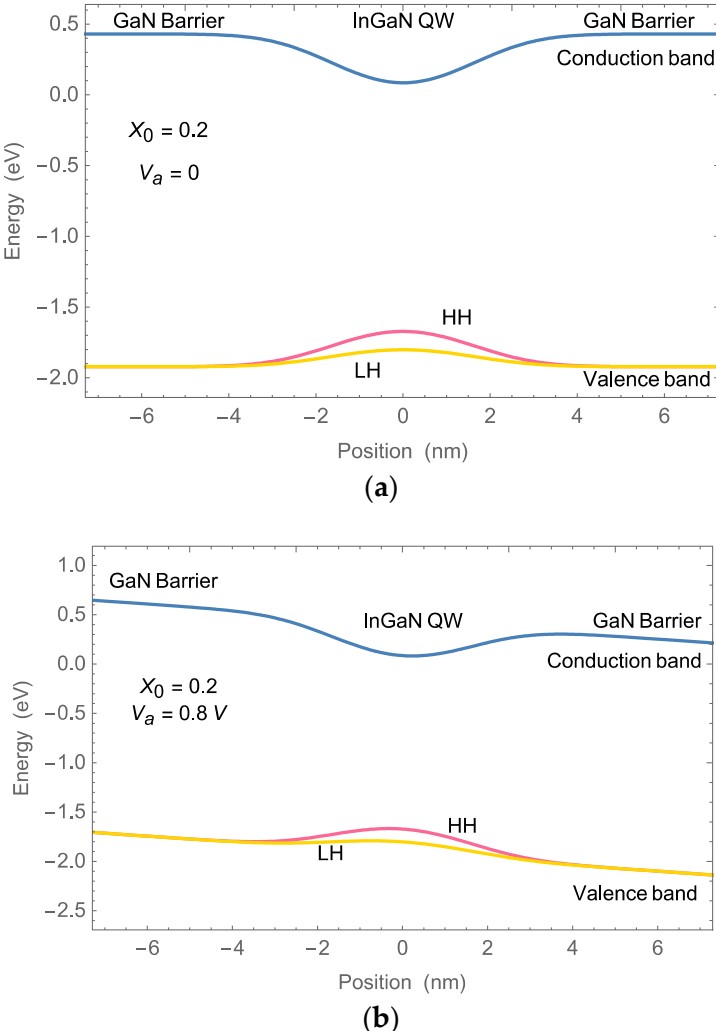

**Figure 2.** Schematic diagram of the In$_x$Ga$_{1-x}$N/GaN parabolic MQW LED structures at an applied bias of (**a**) 0 V and (**b**) 0.8 V.

In Figures 3 and 4, the barrier height $V_0$ was fixed at (0.344, 0.51, and 0.67 eV) in the conduction band, (0.25, 0.4, and 0.47 eV) in the valence band for HH, and (0.12, 0.17, and 0.22 eV) for LH at ($x_0 = 0.2$, 0.3, and 0.4), respectively; the InGaN quantum well width was 3 nm, and the GaN barrier width was 1 nm, and 12 nm, both of which remained fixed in the absence of applied voltage. The strain increased along with the mole fraction. As a result of the increased strain, the conduction and HH bands were pushed upward in the same direction, which led to a decrease in the energy gap. Accordingly, the energy gap values decreased (1.76, 1.6, and 1.4 eV) with the increase in $x_0 = 0.2$, 0.3, and 0.4, respectively.

Figure 5 shows that the LH band was pushed upward in the same direction as the conduction and HH bands, with increased $x_0$ and decreased strain. The energy splitting of HH and LH bands was increased (129.7, 193.7, and 255.7 meV) with the increase in $x_0 = 0.2$, 0.3, and 0.4, respectively.

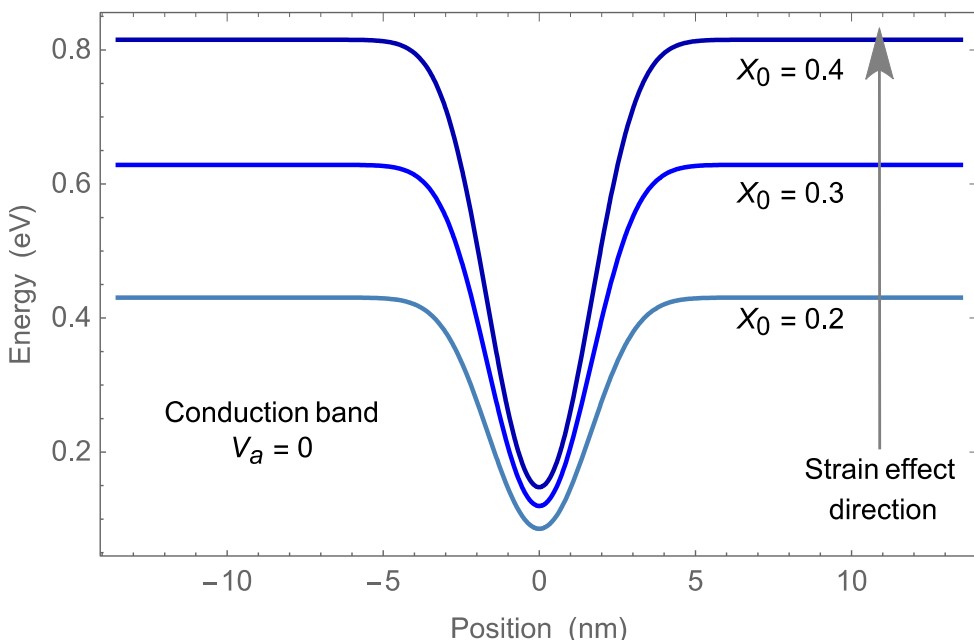

**Figure 3.** Strain effect direction for conduction band of the In$_x$Ga$_{1-x}$N/GaN parabolic LED structures with various mole fractions $(x_0)$ at an applied bias of 0 V.

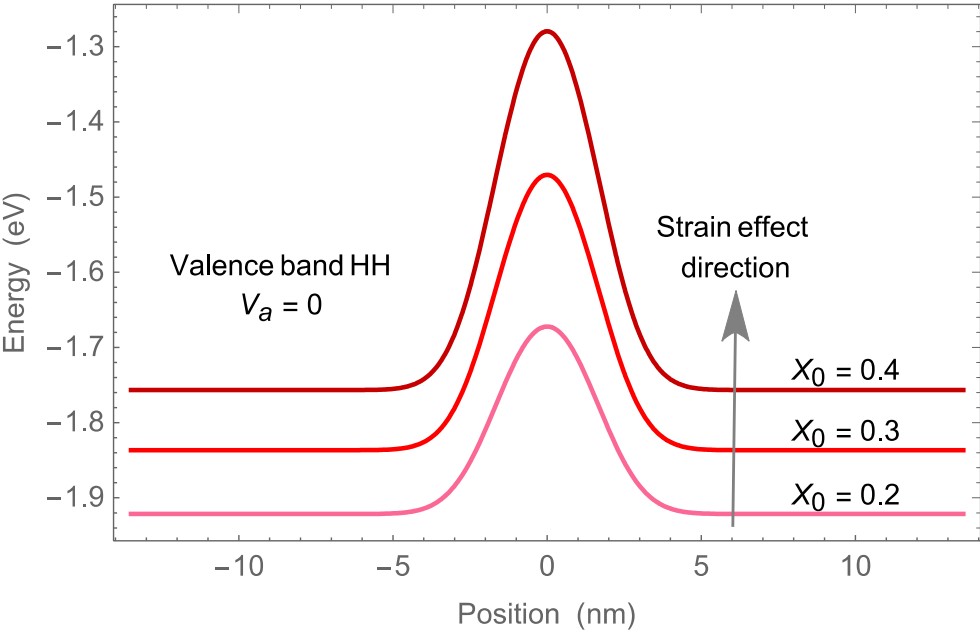

**Figure 4.** Strain effect direction for valence band (HH) of the In$_x$Ga$_{1-x}$N/GaN parabolic LED structures with various mole fractions $(x_0)$ at an applied bias of 0 V.

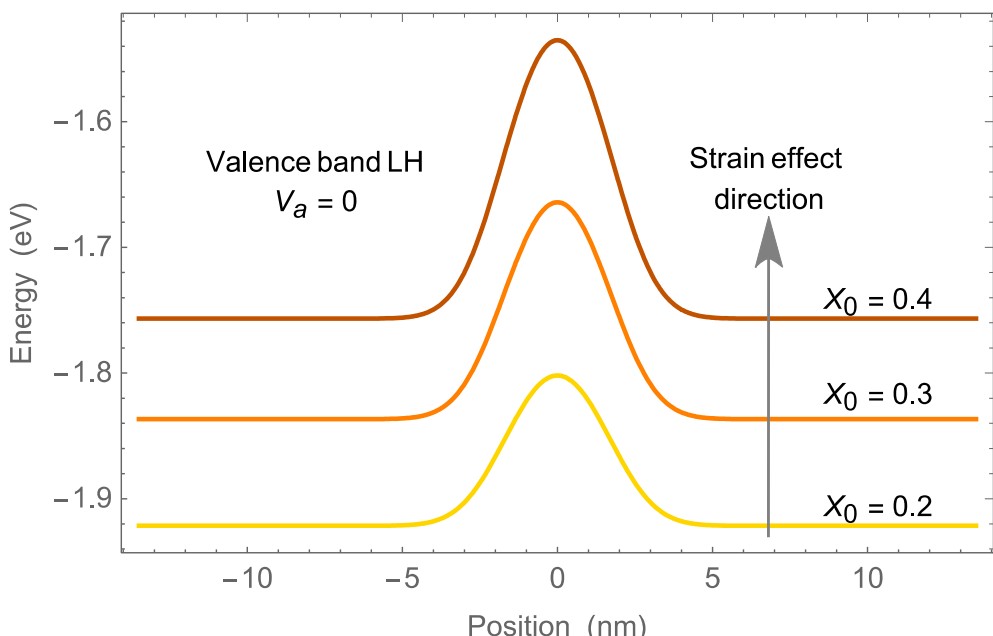

**Figure 5.** Strain effect direction for valence band (LH) of the $In_xGa_{1-x}N/GaN$ parabolic LED structures with various mole fractions $(x_0)$ at an applied bias of 0 V.

Figures 6 and 7 show the logarithm of the transmission coefficient for the electrons and HH as a function of energy for the double-barrier $In_xGa_{1-x}N/GaN$ parabolic quantum well structure with a barrier width = 1 nm, for various values of $x_0$. The calculation for the barrier height $V_0$ was as indicated in Figures 3–5. The electric field extended across the structure at an applied bias of 0.8 V. The transmission coefficient of the electrons in Figure 6 shows single energy peaks at 0.071, 0.17, and 0.25 eV for various $x_0$. In Figure 7, the transmission coefficient of HH shows single energy peaks at 0.188, 0.079, and 0.221 eV for various $x_0$. It is clear that the energy of the electron transmission peak increases along with $x_0$, whereas the energy of the transmission peak of HH fluctuates. The energies presented in Figures 6 and 7 are in a bound state, as they have lower values than the barrier height for various values of $x_0$. The transmission coefficient of the electron is close to unity at $E = 0.85$ eV for $x_0 = 0.2$ and 0.3, and at $E = 1$ eV for $x_0 = 0.4$. Likewise, the transmission coefficient of HH only close to unity at $E = 0.31$ and 0.44 eV for $x_0 = 0.3$ and 0.4, respectively. The tunneling probability of the electrons and HH incident from the GaN side of the InGaN/GaN interface can be determined from these energies $E$. These observations indicate that with increasing $x_0$, the transmission peak of electrons and HH shifted to higher energies, except at $x_0 = 0.2$, in the case of the latter. Under an electric field, electrons and HH exhibit less oscillation and tend to be stable, due to the slope of the barrier height, which explains why only one peak is present.

Figures 8 and 9 show the transmission coefficient of electrons and HH, respectively, at barrier width = 12 nm for various values of $x_0$. The peak number of the transmission coefficient of the electrons and HH increase with increasing barrier widths. The transmission coefficient of electrons in Figure 8 shows three energy peaks, while in Figure 9, the transmission coefficient of HH has more than three energy peaks. At a barrier width of 12 nm, the energy of the electrons and the HH transmission peaks behave similarly to the behavior of those at 1 nm with respect $x_0$. The energies presented in Figures 8 and 9 are in an unbound state, as they have higher values than the barrier height for various values of $x_0$.

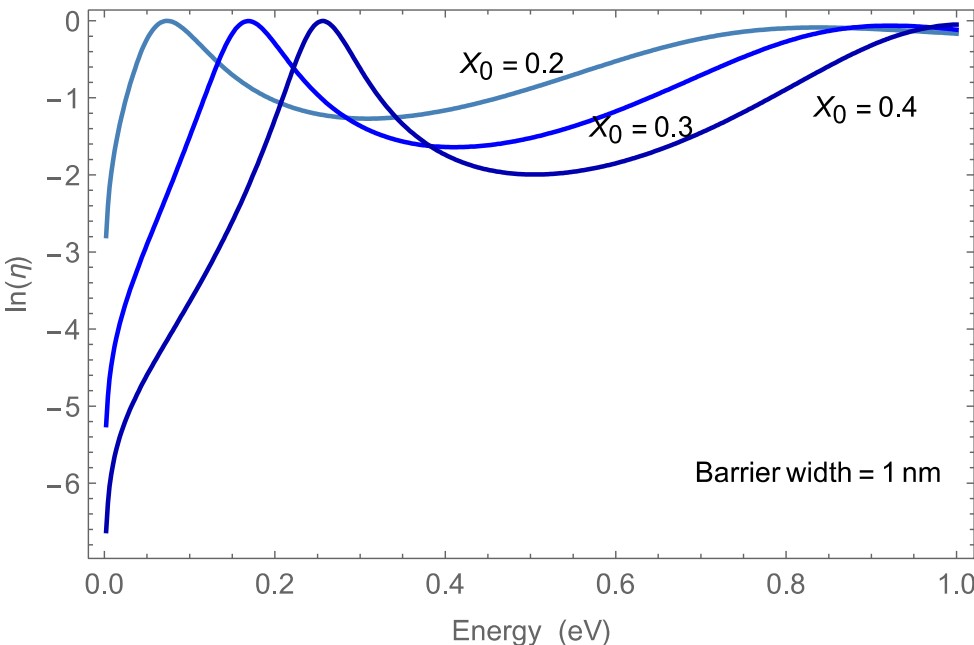

**Figure 6.** Logarithm of the transmission coefficient for electrons as a function of energy for the double-barrier In$_x$Ga$_{1-x}$N/GaN parabolic quantum well structure with various $x_0$ at an applied bias of 0.8 V, and barrier width 1 nm.

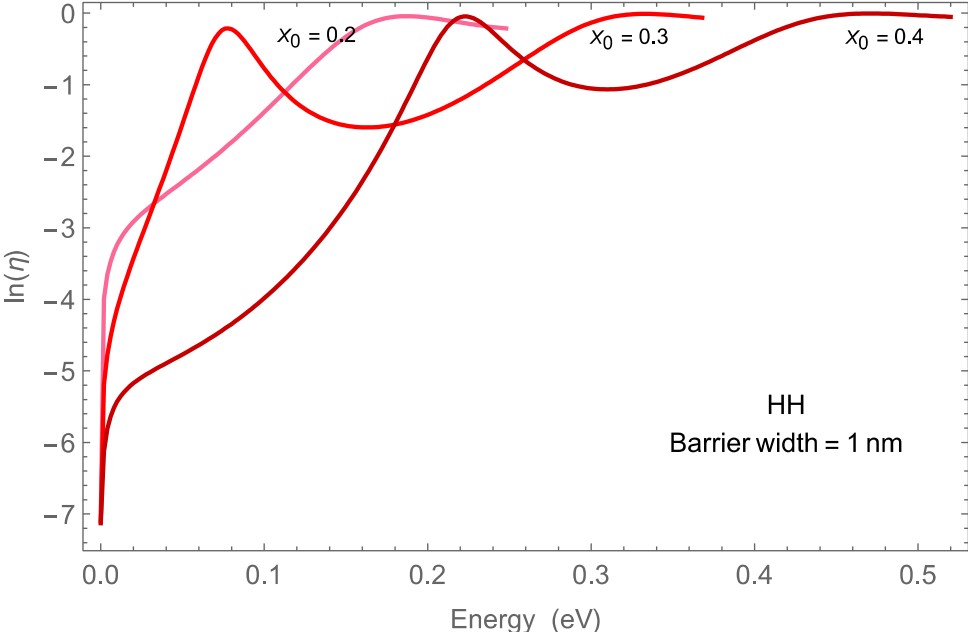

**Figure 7.** Logarithm of the transmission coefficient for HH as a function of energy for the double-barrier In$_x$Ga$_{1-x}$N/GaN parabolic quantum well structure with various $x_0$ at an applied bias of 0.8 V, and barrier width 1 nm.

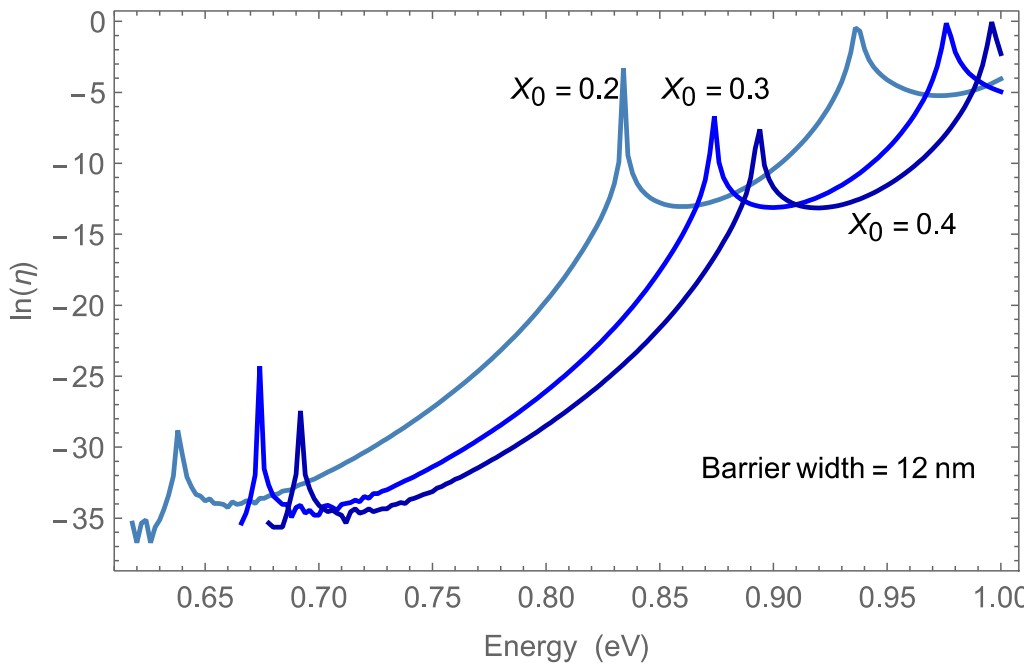

**Figure 8.** Logarithm of the transmission coefficient for electrons as a function of energy for the double-barrier In$_x$Ga$_{1-x}$N/GaN parabolic quantum well structure with various $x_0$ at an applied bias of 0.8 V, and barrier width 12 nm.

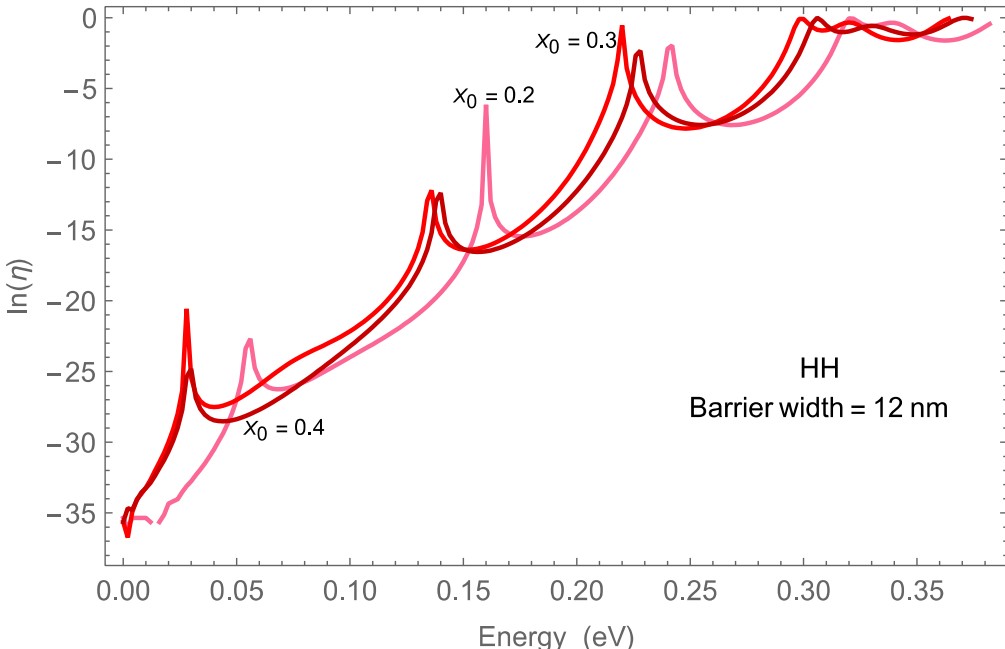

**Figure 9.** Logarithm of the transmission coefficient for HH as a function of energy for the double-barrier In$_x$Ga$_{1-x}$N/GaN parabolic quantum well structure with various $x_0$ at an applied bias of 0.8 V, and barrier width 12 nm.

Figure 10 shows the logarithm of the transmission coefficient for LH as a function of energy for various values of $x_0$. No peaks are observed, indicating the absence of oscillation. The transmission of LH shifts to higher energies with increased $x_0$, mimicking the behavior of the electron state. The transmission is lower for LH than HH. The LH–HH splitting $\Delta_{LH-H} = 0.13$, 0.19, and 0.255 eV increases along with $x_0$, exceeding the Fermi energy

($\approx 0.054$); therefore, only the HH band is dominant in terms of the valence band properties, with no crossing or anti-crossing of the LH and HH bands [6].

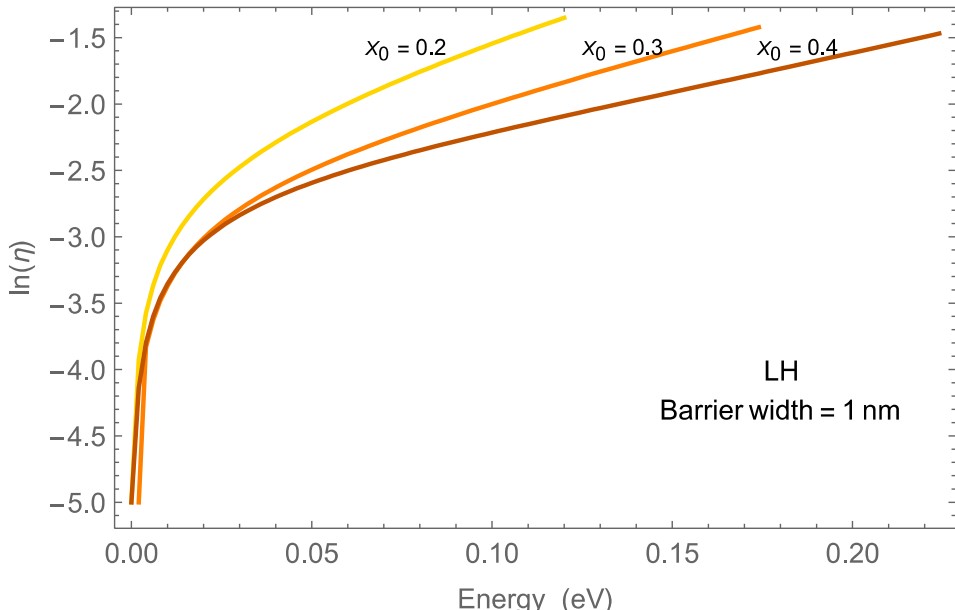

**Figure 10.** Logarithm of the transmission coefficient for LH as a function of energy for the double-barrier $In_xGa_{1-x}N/GaN$ parabolic quantum well structure with various $x_0$ at an applied bias of 0.8 V, and barrier width 1 nm.

Figure 11 shows peaks occurring with the increased barrier width at 12 nm for the transmission of LH, and increasing with increased $x_0$. The transmission is higher for LH at a barrier width = 12 nm than for LH at 1 nm.

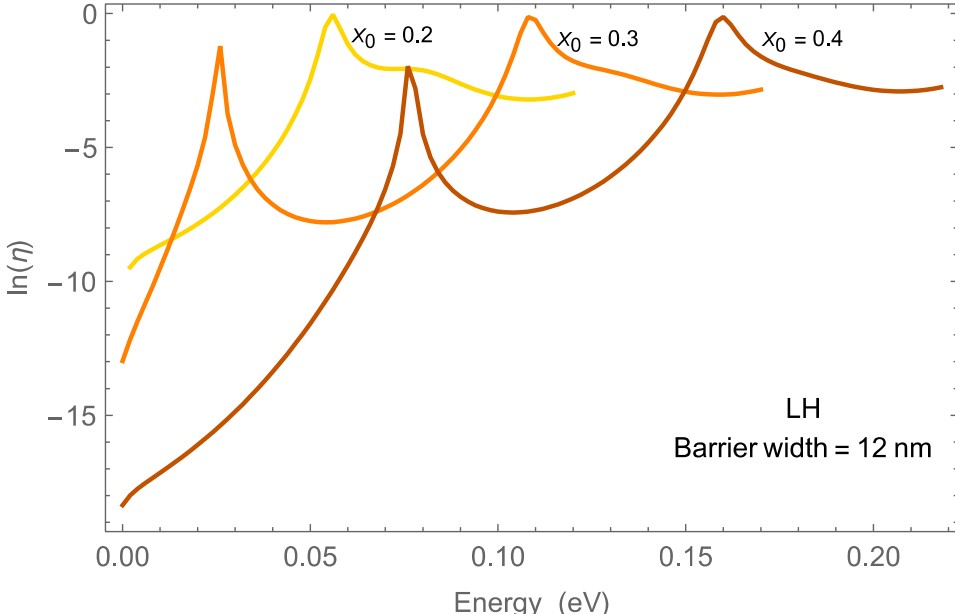

**Figure 11.** Logarithm of the transmission coefficient for LH as a function of energy for the double-barrier $In_xGa_{1-x}N/GaN$ parabolic quantum well structure with various $x_0$ at an applied bias of 0.8 V, and barrier width 12 nm.

According to the calculations in Figures 6–10, the tunneling lifetime ($\tau = \frac{\hbar}{\Gamma_{FWHM}}$) of the electron state decreases with increasing $x_0$ for electrons ($\tau = 10.8, 8.9,$ and $7.5$ fs), but increases for HH ($\tau = 0.41, 0.46, 0.53$ fs), where $\Gamma_{FWHM}$ is the full width at the half maximum of the transmission peak. The tunneling lifetime of the state is the time between transmission events [27]. However, the electron state has a longer tunneling lifetime than the HH state. The tunneling lifetime depends on the width of both the quantum barrier and the well, as well as the external electric field. Strongly electron bound states have a longer tunneling lifetime. Resonant tunneling devices eventually reach their speed limits due to the tunneling lifetime [34]. The probability of LH tunneling was about 22.3%, which is smaller than that of electron and HH tunneling. This may be explained by the small barrier height $V_0$ for the LH band, potentially indicating that all the states were bound states with finite tunneling lifetimes $\tau$. The lifetimes $\tau$ increased with an increase in barrier width, reaching up to 65 fs at 12 nm.

The relationships between the current density and applied voltages of 1 nm, and 12 nm for the barrier thickness for various values of $x_0$, are illustrated in Figures 12–17. The current density peaks that occurred under the conditions of an electric field that has been applied across the InGaN/GaN structure is evidence of electron resonant tunneling through the potential well [4,15,27]. A comparison between the electron tunneling current density and HH tunneling current density revealed that the current density peaks for electrons tended to occur at higher voltages with increased $x_0$ than for the HH at a barrier width of 1 nm. The electron tunneling current density is decreased with increased $x_0$ at a barrier width of 1 nm. The number of current density peaks increased for the electrons and HH at a barrier width of 12 nm. The electron tunneling current density is higher than HH and LH tunneling current density at a barrier width of 1 nm, while being less than both HH and LH at a barrier width of 12 nm. Additionally, the barrier width = 12 nm had a higher peak-to-valley ratio (PVR) than the barrier width = 1 nm; this indicates a series of NDRs, hence low power dissipation. The NDR property is critical in circuit implementation, because it can provide different voltage-controlled logic states for the peak and valley currents [4,35]. In Figures 16 and 17 no peaks are observed, indicating the absence of NDRs. The LH tunneling current density decreased with increased $x_0$.

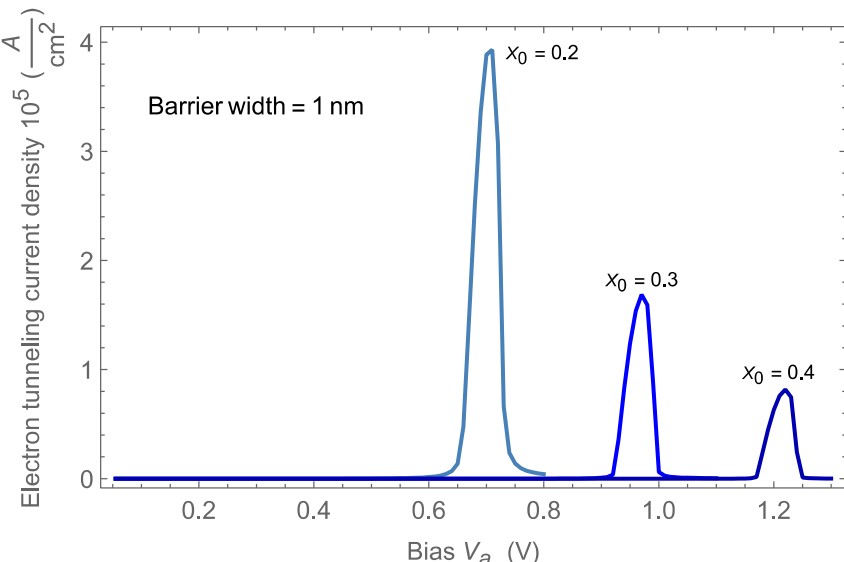

**Figure 12.** Current density-voltage characteristics of the electron tunneling at width barrier 1 nm.

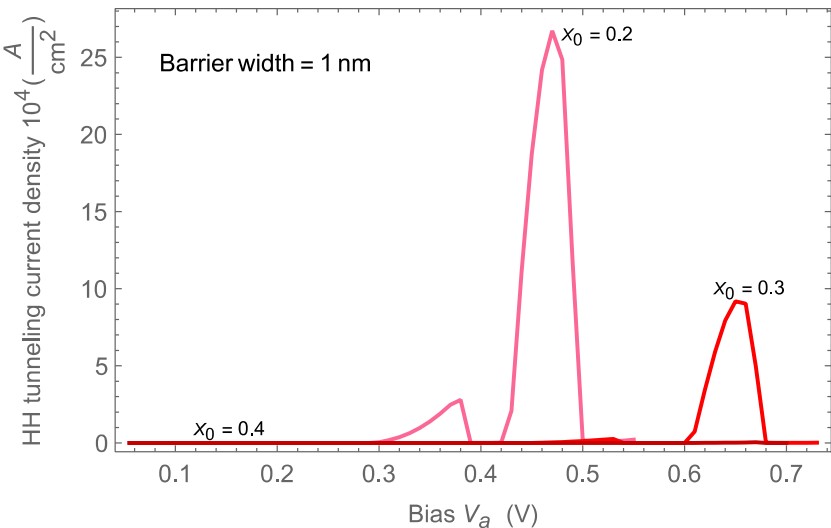

**Figure 13.** Current density-voltage characteristics of the HH tunneling at width barrier 1 nm.

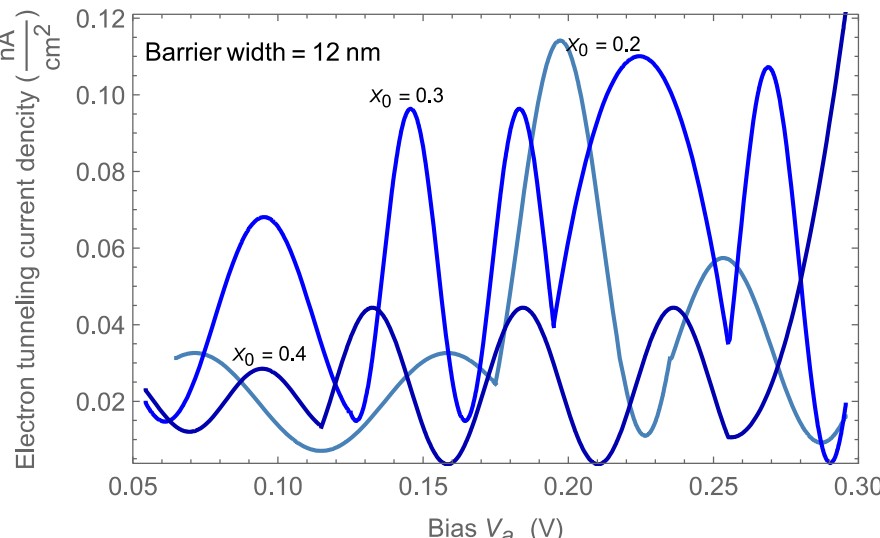

**Figure 14.** Current density-voltage characteristics of the electron tunneling at width barrier 12 nm.

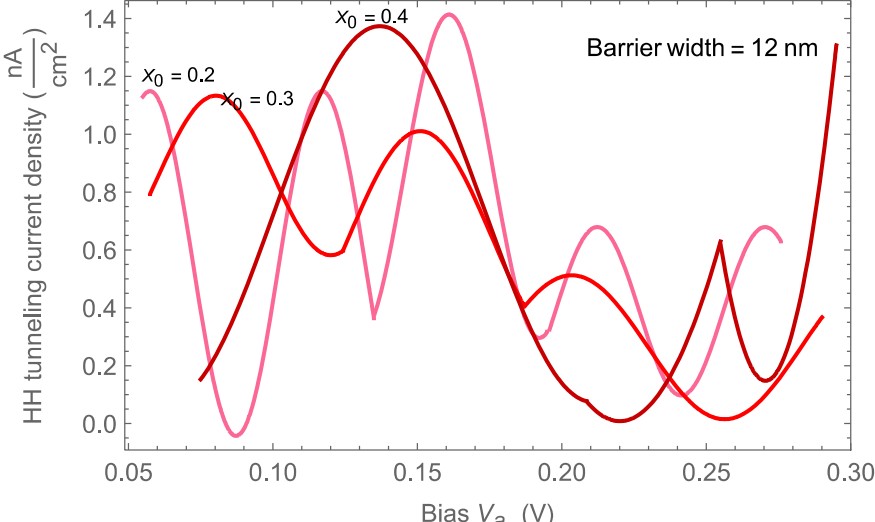

**Figure 15.** Current density-voltage characteristics of the HH tunneling at width barrier 12 nm.

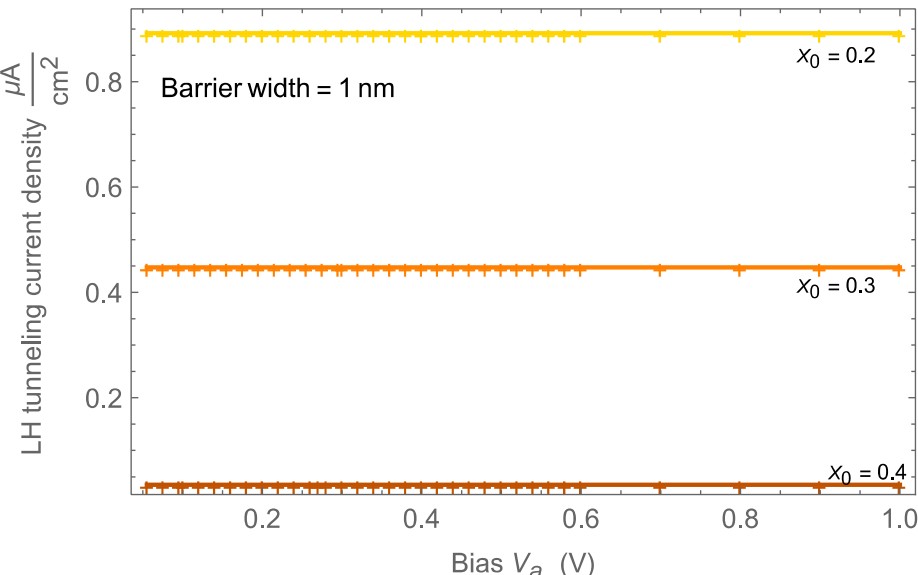

**Figure 16.** Current density-voltage characteristics of the LH tunneling at width barrier 1 nm.

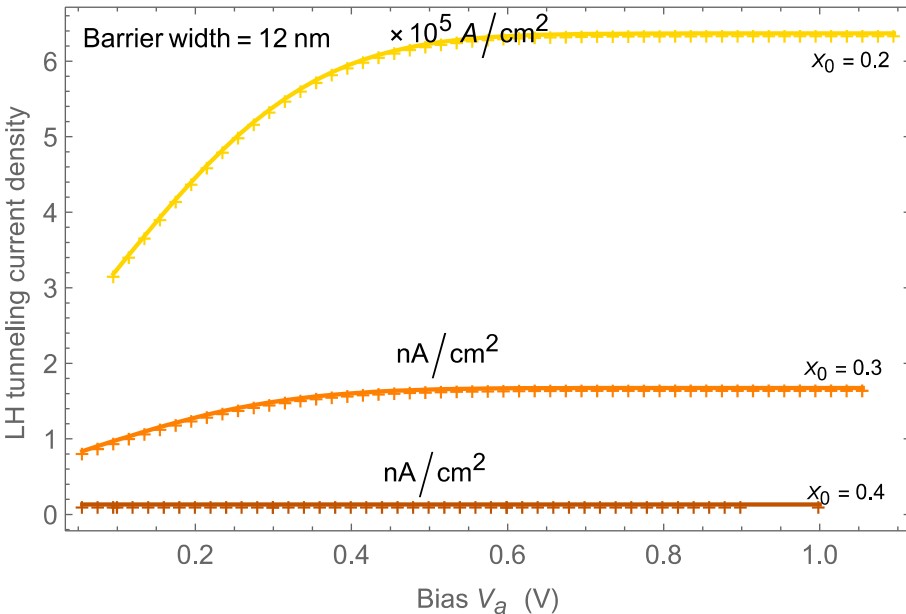

**Figure 17.** Current density-voltage characteristics of the LH tunneling at width barrier 12 nm.

The relationship between the transmission coefficient T(E) and the electric field intensity of the structure was also investigated; the results were used to calculate the current density versus voltage (J–V) characteristics of the resonant tunneling diodes (RTDs) with a negative differential resistance (NDR).

IQE curves as a function of current are plotted in Figures 18 and 19, according to the ABC model in Equation (12). The current values were determined using Equation (11). The calculation shows that an increase in $x_0$ would effectively decrease the efficiency. The efficiency droop at a current of 1 A would be 3%, 14%, and 65% at a barrier width of 1 nm, and would be 4%, 20%, and 75% at a barrier width of 12 nm, with decreasing $x_0$. The reason for this efficiency droop could be related to the increase in current and decrease in tunneling lifetime. A decrease in tunneling lifetime would reduce the escape speed of particles. That means the LED with the thicker quantum barriers has a smaller efficiency droop. Therefore, we obtained a slight amelioration with increased barrier width.

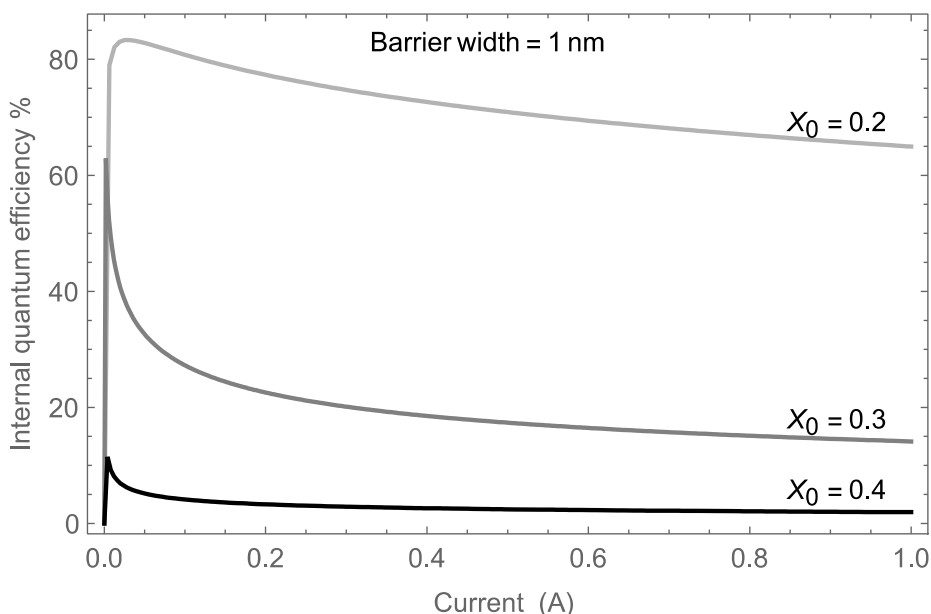

**Figure 18.** Internal quantum efficiency as a function of current at barrier width 1 nm.

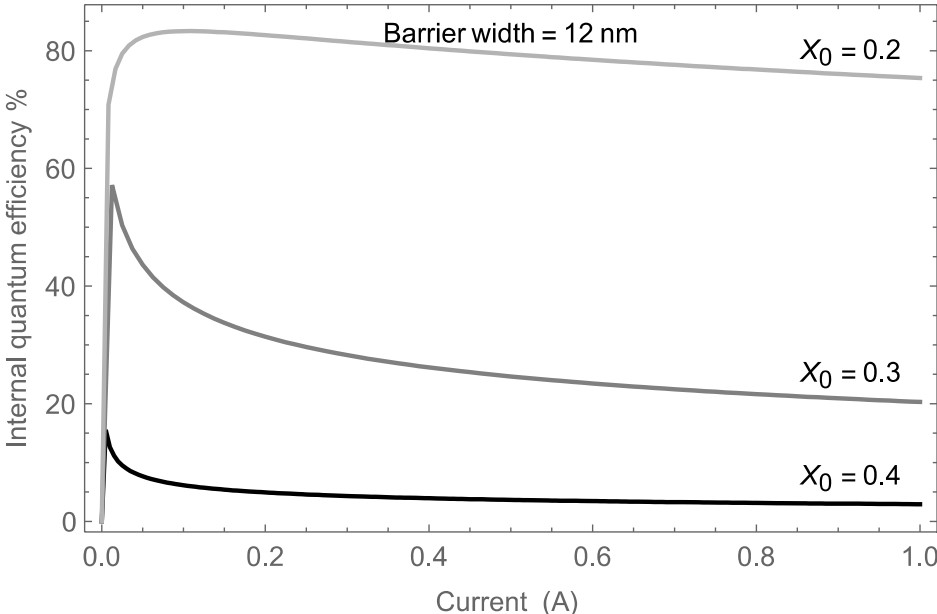

**Figure 19.** Internal quantum efficiency as a function of current at barrier width 12 nm.

Figure 20 shows normalized internal quantum efficiency as a function of current density at barrier width = 12 nm and $x_0 = 0.2$. The results are obtained from the experimental data [16] and the theoretical results. It is clear that the data obtained from the theoretical results yield results close to the experimental data.

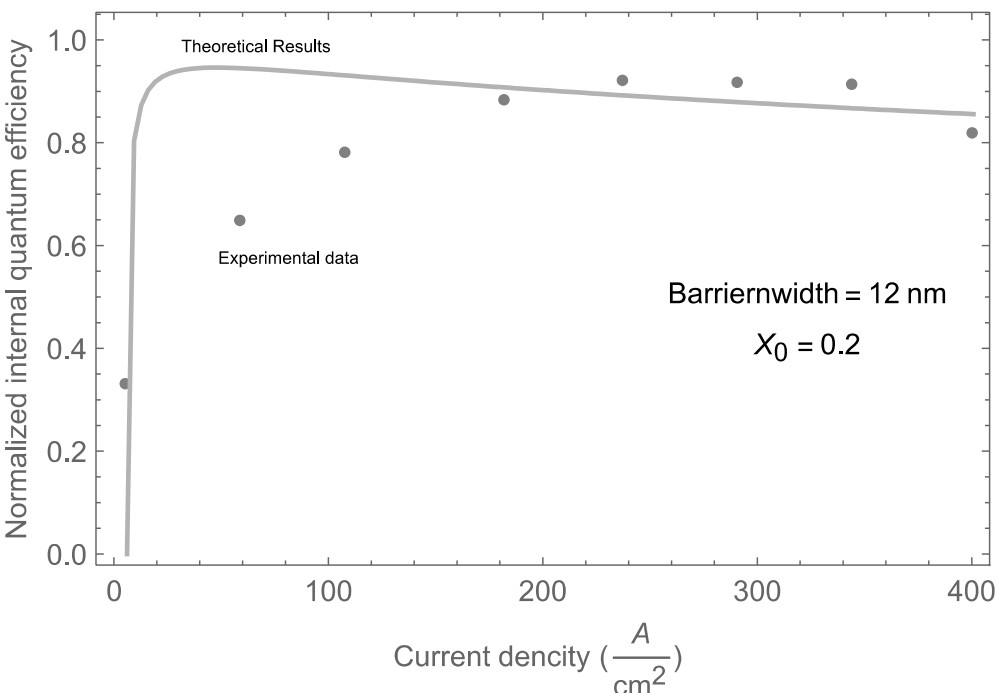

**Figure 20.** Normalized internal quantum efficiency as a function of current density obtained from the previous study [16], along with the theoretical results.

## 4. Conclusions

The effect of the mole fraction on the resonant tunneling of electrons, heavy holes, and light holes through the $In_xGa_{1-x}N/GaN$ parabolic quantum well/LED structure, as well as its efficiency, was investigated theoretically. In this study, the magnitude of strain and its type were taken into consideration. The increase in strain with the mole fraction pushed the conduction and HH bands upward in the same direction, resulting in a decrease in the energy gap. The results showed that an increase in the mole fraction tended to decrease the tunneling lifetime of electrons, although the barrier height increased, while the tunneling lifetime of HH increased. Moreover, the transmission peaks of electrons and HH shifted to a higher energy with the increase in the mole fraction. The LH–HH splitting value increased with the mole fraction and remained higher than the Fermi energy; therefore, only the HH band was dominant in terms of the valence band properties, with no crossing or anti-crossing of the LH and HH bands. The calculations showed that an increase in the mole fraction decreases the efficiency. However, the difference between barrier width 1 nm and 12 nm is minimal in the efficiency droop, except that the efficiency can be increased by increasing the quantum barrier width and taking into account the decreased mole fraction. The series of NDRs in the thicker barrier indicate low power dissipation.

**Funding:** This work was supported by Imam Abdulrahman Bin Faisal University, Dammam, Saudi Arabia.

**Institutional Review Board Statement:** Not applicable.

**Informed Consent Statement:** Not applicable.

**Data Availability Statement:** Not applicable.

**Conflicts of Interest:** The authors declare no conflict of interest.

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
