# Peer review of "Resonant Tunneling of Electrons and Holes through the InxGa1−xN/GaN Parabolic Quantum Well/LED Structure"

_crystals, doi:10.3390/cryst12081166_

Round 1
Reviewer 1 Report
I carefully read the manusscript entitled "Resonant Tunneling of Electrons and Holes through InGaN/GaN Parabolic Quantum Well/LED Structure". The manusscript is well written and clearly structured. It is also clear that the study has a significance for real world applications. There are a few points that I think need to be added to the manusscript. At the moment, I can only recommend the paper for publication, if the points are addressed adequatly. Below, please find a few comments/questions/concerns:
1. The end of the first part of the introduction is focused on resonant tunneling in GaAs/AlGaAs. However, there are also a lot activities of RTDs based on III-N, the author does not discuss them at all. Especially, the long endavor to observe resonant tunneling in these structures. The author does also not discuss the issue of RTDs on III-N that are mainly caused by the large internal fields due to the large piezo-electric constants. This is however very ciritical. While one can calculate the transmission function and see e.g. bound states, that does not imply that resonant tunneling will occur in a real device.
2. The author presents that the transmission reaches unity even under an applied electric field. I would not expect to have T=1 for such a case, as the two tunneling barriers are not identical anymore. One would expect T=1, only for symmetric structures and the field breaks this symmetry. So, why is T=1?
3. The author uses barrier thicknesses of a few nm (in another publication up to 12nm). I would expect that the tunneling current for such thick barriers is very low, much lower than the threshold current density. Thus, I would assume that the resonant tunneling contribution is very low, compared to other current contributions.
4. The author should check that all acronyms are well defined, e.g. MBE.
Author Response
"Please see the attachment."

Reviewer 2 Report
In this manuscript, the authors developed models describing the tunneling of electrons and holes through parabolic InxGaN1−x/GaN quantum well/LED structures with respect to strain. In my opinion, this manuscript is interesting to the readers of Crystals. The topic is very important in this field. This work is novel and original. The authors have solid background in this field. Therefore, the referee recommends it to be published after the following revisions:
1. The English should be polished by a native speaker.
2. It is suggested the authors to cite more recent reports (2020~2022).
3. The introduction part should be divided into several paragraphs.
4. The term of the compounds needs to be subscripted (e.g. In0.2Ga0.8N)
4. The referee suggests enriching the comparison with the state-of-the-art using different simulation methods or models, which will be very important for the readers in this field.
In general, this work seems to be very interesting. The referee would like to see the revision if possible.
Author Response
"Please see the attachment."
